# Large Language Models are Temporal and Causal Reasoners for Video Question Answering

**Dohwan Ko**[1*] **Ji Soo Lee**[1*] **Wooyoung Kang**[2] **Byungseok Roh**[2] **Hyunwoo J. Kim**[1†]

[1]Department of Computer Science and Engineering, Korea University [2]Kakao Brain

{ikodoh, simplewhite9, hyunwoojkim}@korea.ac.kr

{edwin.kang, peter.roh}@kakaobrain.com

## Abstract

Large Language Models (LLMs) have shown remarkable performances on a wide range of natural language understanding and generation tasks. We observe that the LLMs provide effective priors in exploiting *linguistic shortcuts* for temporal and causal reasoning in Video Question Answering (VideoQA). However, such priors often cause suboptimal results on VideoQA by leading the model to over-rely on questions, *i.e.*, *linguistic bias*, while ignoring visual content. This is also known as 'ungrounded guesses' or 'hallucinations'. To address this problem while leveraging LLMs' prior on VideoQA, we propose a novel framework, Flipped-VQA, encouraging the model to predict all the combinations of ⟨V, Q, A⟩ triplet by flipping the source pair and the target label to understand their complex relationships, *i.e.*, predict A, Q, and V given a VQ, VA, and QA pairs, respectively. In this paper, we develop LLaMA-VQA by applying Flipped-VQA to LLaMA, and it outperforms both LLMs-based and non-LLMs-based models on five challenging VideoQA benchmarks. Furthermore, our Flipped-VQA is a general framework that is applicable to various LLMs (OPT and GPT-J) and consistently improves their performances. We empirically demonstrate that Flipped-VQA not only enhances the exploitation of linguistic shortcuts but also mitigates the linguistic bias, which causes incorrect answers over-relying on the question. Code is available at `https://github.com/mlvlab/Flipped-VQA`.

## 1 Introduction

Large Language Models (LLMs) have exhibited an impressive ability to free-form generation tasks and multi-choice question-answering tasks in natural language processing (Chung et al., 2023; Jiang et al., 2023; Fei et al., 2023; Cai et al., 2022; Chen et al., 2023; Saha et al., 2022). These LLMs have

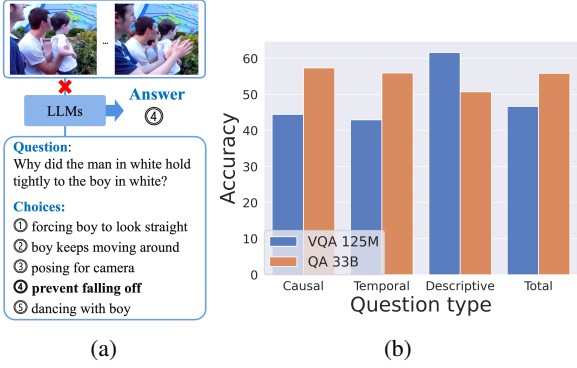

Figure 1: **LLMs' temporal and causal reasoning ability.** (a) An example of a causal question that LLMs correctly answer without visual content. (b) Comparison of LLaMA 33B (QA) and OPT 125M (VQA).

achieved human-level performance on a wide range of challenging tasks like professional & academic QA (Hendrycks et al., 2021), science QA (Clark et al., 2018), mathematics QA (Cobbe et al., 2021), code generation (Chen et al., 2021), and commonsense reasoning (Zellers et al., 2019; Sakaguchi et al., 2021) since they are pretrained with large-scale corpora (*e.g.*, CommonCrawl, Bookcorpus, and Wikipedia) which entail massive human knowledge. With such pretraining data, usually comprising a series of contexts, LLMs are trained to predict the next token given the preceding tokens. Therefore, LLMs learn to predict the next context given a series of contexts during pretraining, so they implicitly learn *temporal* and *causal* reasoning ability.

To assess LLMs' temporal and causal reasoning ability, we explore a popular multi-modal understanding task, Video Question Answering (VideoQA), which requires the model to predict the correct answer (A) given a video (V) and question (Q) pair. Recent challenging VideoQA benchmarks demand the model to answer the question which asks temporal and causal relationships, *e.g.*, the next event of a video or the reason why a scene happens. We observe that LLMs effectively handle

---
[*]Equal contribution.

[†]Corresponding author.

such challenging VideoQA benchmarks by leveraging their strong prior knowledge of temporal and causal reasoning learned from the pretraining phase. For example, in Fig. 1a, LLMs correctly answer causal questions solely based on the text question and options without referring to the visual content by exploiting *linguistic shortcut*. Also, Fig. 1b shows that a language-only QA model equipped with a larger language model, *e.g.*, LLaMA 33B, denoted by QA 33B outperforms a VideoQA model with OPT 125M trained with full ⟨V, Q, A⟩ on causal and temporal questions by a large margin of 13%. Although LLMs' prior knowledge is effective for addressing complex temporal and causal questions, this sometimes leads to suboptimal answers when the model overly depends on inaccurate linguistic prior, *i.e.*, *linguistic bias*, while ignoring the visual content. This is known as the 'hallucination problem' of visual question-answering models equipped with LLMs (Alayrac et al., 2022). Although in the literature linguistic shortcut and linguistic bias are interchangeably used, in this paper, we use the former specifically when the linguistic prior is correct and the latter otherwise.

Here, we propose a novel learning framework, Flipped-VQA, predicting all the combinations of ⟨V, Q, A⟩ triplet by flipping the source pair and the target label, *i.e.*, VQ → A (main task), VA → Q, and QA → V (auxiliary tasks). In other words, to understand complex relationships between the video, question, and answer, LLMs are asked to additionally predict the question given a video-answer pair and the video given a question-answer pair by leveraging their knowledge of temporal and causal reasoning. In our experiments, we develop LLaMA-VQA by applying Flipped-VQA to LLaMA (Touvron et al., 2023) and it outperforms other baselines on five challenging VideoQA benchmark datasets: NExT-QA (Xiao et al., 2021), STAR (Wu et al., 2021), DramaQA (Choi et al., 2021), VLEP (Lei et al., 2020), and TVQA (Lei et al., 2018) with a small number of learnable parameters (only 0.06% of total model parameters). Furthermore, Flipped-VQA improves the performance of GPT-J (Wang and Komatsuzaki, 2021) and OPT (Zhang et al., 2022), implying that our framework is generally applicable to other decoder-only LLMs. We empirically demonstrate that Flipped-VQA encourages LLMs to exploit linguistic shortcuts by leveraging their prior knowledge and also mitigates linguistic bias which causes

incorrect answer over-relying on the question. To sum up, our **contributions** are as follows:

- We investigate that pretrained LLMs' knowledge is a strong prior for temporal and causal reasoning on challenging VideoQA.

- We propose a novel framework, Flipped-VQA, to efficiently fine-tune LLMs on VideoQA by reasoning and understanding the complex relationships of ⟨V, Q, A⟩ triplet, using LLMs' prior knowledge of temporal and causal reasoning. Flipped-VQA requires LLMs to perform three tasks: VQ → A, VA → Q, and QA → V, and we combine these objectives as LLMs' language generation objective.

- LLaMA-VQA trained by Flipped-VQA, outperforms the baselines on five challenging VideoQA benchmark datasets. Also, our experiments demonstrate that Flipped-VQA is generally applicable to various decoder-only LLMs and consistently improves their performances.

- Our extensive analyses show that Flipped-VQA is effective in exploiting linguistic shortcuts to answer the question based on LLMs' prior knowledge and alleviating the linguistic bias by increasing the utilization of visual contents.

## 2 Related works

**LLMs for temporal and causal reasoning.** Exposed to a wide range of corpora during pretraining, LLMs perform diverse reasoning tasks (Liu et al., 2023; Wang et al., 2023c; Ozturkler et al., 2023; Ho et al., 2022; Li et al., 2022; Wang et al., 2023c; Kıcıman et al., 2023). Particularly, a line of works (Zhang et al., 2023a; Li et al., 2023e; Kamalloo et al., 2023; Wang et al., 2023b; Tan et al., 2023) focuses on the temporal and causal reasoning skills of LLMs. For temporal reasoning, Zhang and Choi (2021) assesses LLMs by asking open-domain time-sensitive questions in both closed and open settings. Similarly, Hobbhahn et al. (2022) investigates the ability through querying subject and relation pairs of different time periods. Furthermore, some works have also explored the causal capabilities by evaluating whether LLMs can understand the causal implications given in the sentence. Long et al. (2023) uses simple causal graphs and determines if LLMs can understand the relationship

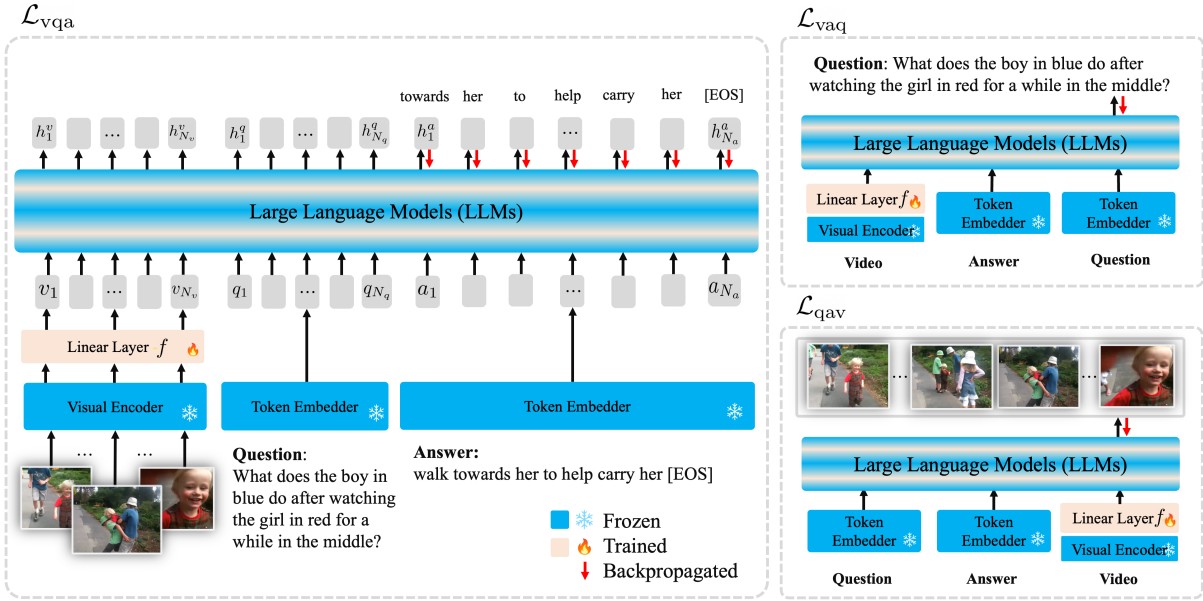

Figure 2: **Illustration of LLMs with Flipped-VQA.** Flipped-VQA consists of three objectives: $\mathcal{L}_{\text{vqa}}$, $\mathcal{L}_{\text{vaq}}$, and $\mathcal{L}_{\text{qav}}$. $\mathcal{L}_{\text{vqa}}$ is a common objective, which predicts the answer given a video-question pair, for VideoQA. Likewise, $\mathcal{L}_{\text{vaq}}$ and $\mathcal{L}_{\text{qav}}$ are the objectives for question and video prediction by leveraging LLMs' knowledge. In other words, for each objective, VQ, VA, and QA pair is used as prefix tokens to predict A, Q, and V, respectively. Trainable parameters interleaved in LLMs stand for adapter tokens as in LLaMA-Adapter. Our framework employs only a relatively small number of trainable parameters on LLMs, *e.g.*, 4.5M trainable parameters among the total parameters of LLaMA 7B (0.06%).

between nodes. In this work, we further examine the temporal and causal reasoning skills of LLMs expanded to the multi-modal setting of challenging VideoQA.

**LLMs for multi-modal understanding.** Various lines of work attempt to incorporate different modalities into LLMs to leverage the models' generation power and knowledge in performing multi-modal downstream tasks. There exist various techniques (Hu et al., 2021; Jia et al., 2022; Ju et al., 2022) in the literature to bridge the gap between different modalities. For instance, Flamingo (Alayrac et al., 2022) ingests visual content into the frozen Chinchilla (Hoffmann et al., 2022) through a Perceiver Resampler. LLaMA-Adapter (Zhang et al., 2023c) fine-tunes LLaMA by applying linear projection along with the adaption of prompts to incorporate the visual information. Recently, there have been several approaches to develop a LLMs-based video chat model trained on massive multi-modal instruction tuning dataset that enables comprehensive understanding across different modalities (Zhang et al., 2023b; Li et al., 2023b,d). Specifically, VideoChat (Zhang et al., 2023b) proposes video-centric instruction dataset that primarily emphasizes spatio-temporal reasoning and causal rela-

tionships present in the video.

**Video Question Answering (VideoQA).** VideoQA aims to answer natural language questions given a video that requires multi-modal understanding and reasoning skills on different semantic levels. Previous VideoQA benchmarks (Xu et al., 2017; Jang et al., 2017) target short videos and ask questions based on visual facts such as location and objects/attributes. In contrast, more recent benchmarks (Xiao et al., 2021; Lei et al., 2020, 2018; Wu et al., 2021; Choi et al., 2021) tend to tackle temporal and causal questions referencing a longer video. Specifically, NExT-QA (Xiao et al., 2021) requires uncovering the cause/intention of a certain event *(e.g., Why did ...)* or reasoning about subsequent actions in the video *(e.g., What/How ... do/react after ...)*[1]. In this work, we address these challenging VideoQA benchmarks through LLMs' temporal and causal reasoning abilities.

## 3 Method

We present Flipped-VQA, a simple yet effective framework for Video Question Answering (VideoQA) that leverages the LLMs' prior knowl-

---

[1]Further details with examples of NExT-QA are provided in Sec. D.

edge of temporal and causal reasoning. In addition to the target task of predicting an answer A given a video V and a question Q (*i.e.*, VQ → A), our framework flips the role of inputs and outputs, requiring the model to predict V given QA and Q given VA. We apply our framework to LLaMA (Touvron et al., 2023) and develop LLaMA-VQA but it is applicable to any decoder-only LLMs. In this section, we first describe the overall architecture of LLaMA-VQA and then introduce our objective. The overall architecture is illustrated in Fig. 2.

## 3.1 LLaMA-VQA

LLaMA-VQA is built on LLaMA with a few additional learnable parameters. First, LLaMA-VQA adopts a learnable linear layer $f$ to project the visual embeddings, extracted from the frozen visual encoder CLIP ViT/L14 (Radford et al., 2021), to LLaMA's text token embedding space, see Fig. 2. Specifically, given a raw video $x_v$, a sequence of visual tokens is calculated as $\mathbf{v} = [v_1, \ldots, v_{N_v}] = f(\text{CLIP}(x_v)) \in \mathbb{R}^{N_v \times D}$, where $N_v$ is the number of video frames and $D$ is a feature dimension. Second, as in LLaMA-Adapter (Zhang et al., 2023c), we additionally adopt several trainable adapter tokens $\mathbf{p} = [p_1, \ldots, p_{N_p}]$ which are prepended to the key and value of each self-attention layer, where $N_p$ is the number of adapter tokens. Further descriptions of LLaMA-Adapter are provided in Sec. C. So the number of trainable parameters of LLaMA-VQA 7B is 4.5M, only 0.06% of total parameters of LLaMA 7B. With such a few trainable parameters, LLaMA-VQA effectively preserves LLMs' prior knowledge and leverages it in exploiting linguistic shortcuts for VideoQA.

The question $\mathbf{q} = [q_1, \ldots, q_{N_q}] \in \mathbb{R}^{N_q \times D}$ and answer $\mathbf{a} = [a_1, \ldots, a_{N_a}] \in \mathbb{R}^{N_a \times D}$ tokens are extracted from raw question $x_q$ and answer $x_a$ texts by a tokenizer, where $N_q$ and $N_a$ are the numbers of question and answer tokens respectively. The input prompt with visual tokens for LLaMA-VQA is provided in Tab. 1, where $\mathbf{v}$ and $\mathbf{q}$ serve as prefix tokens, see Sec. B for further prompt details. For simplicity, we omit the tokens for the prompt template (*e.g.*, 'Video:' and 'Question:') and only consider content tokens (*e.g.*, '<$v_1$>' and '<question>') in our equations. Note that $\mathbf{q} \in \mathbb{R}^{N_q \times D}$ represents only the question tokens and choice tokens are omitted in following notations. Then, token sequences $\mathbf{v}$, $\mathbf{q}$, and $\mathbf{a}$ are concatenated and fed to

```
[SOS] Video: <v₁><v₂> ··· <v_Nᵥ>
Question: <question>
Choices:
(A) <option 1>
(B) <option 2>
(C) <option 3>
(D) <option 4>
(E) <option 5>
Answer: The answer is <answer> [EOS]
```

Table 1: **Input Prompt of LLaMA-VQA.**

LLaMA, and the output feature is calculated as:

$$[\mathbf{h}^v, \mathbf{h}^q, \mathbf{h}^a] = \text{LLaMA}([\mathbf{v}, \mathbf{q}, \mathbf{a}], \mathbf{p}), \quad (1)$$

where $\mathbf{h}^v$ is a sequence of output features, *i.e.*, $\mathbf{h}^v = [h_1^v, \ldots, h_{N_v}^v]$, and $\mathbf{h}^q, \mathbf{h}^a$ are similarly defined.

## 3.2 Flipped-VQA

To utilize LLMs' temporal and causal reasoning abilities, we here present Flipped-VQA, consisting of three objectives, for reasoning the complex relationship between video, question, and answer of VideoQA.

**VQ → A.** Predicting an answer given a video-question pair is the primary task of VideoQA. Its objective function is formulated as:

$$\mathcal{L}_{\text{vqa}} = -\log P(\mathbf{a}|\mathbf{v}, \mathbf{q})$$
$$= -\sum_{t=0}^{N_a-1} \log P(a_{t+1}|\mathbf{v}, \mathbf{q}, a_{\leq t}), \quad (2)$$

where $\mathbf{v}$ and $\mathbf{q}$ are given as prefix to generate the answer $\mathbf{a}$. Note that $P(a_1|\mathbf{v}, \mathbf{q}, a_{\leq 0}) := P(a_1|\mathbf{v}, \mathbf{q})$. Then, the probability in Eq. (2) is calculated as:

$$P(a_{t+1}|\mathbf{v}, \mathbf{q}, a_{\leq t}) = \text{Softmax}(\text{Linear}(h_t^a)). \quad (3)$$

At the inference phase, the model predicts the answer as:

$$\hat{\mathbf{a}} = \underset{\mathbf{a} \in \mathcal{A}}{\arg\max} \, P(\mathbf{a}|\mathbf{v}, \mathbf{q}), \quad (4)$$

where $\mathcal{A}$ is a set of candidate answers, *i.e.*, choices.

We now flip the role of inputs and outputs and define two auxiliary tasks: question generation and video prediction.

**VA → Q.** Similar to $\mathcal{L}_{\text{vqa}}$, we also encourage the model to generate the question from the video and

answer as:

$$\mathcal{L}_{\text{vaq}} = - \log P(\mathbf{q}|\mathbf{v}, \mathbf{a})$$
$$= - \sum_{t=0}^{N_q-1} \log P(q_{t+1}|\mathbf{v}, \mathbf{a}, q_{\leq t}), \quad (5)$$

where $P(q_{t+1}|\mathbf{v}, \mathbf{a}, q_{\leq t}) = \text{Softmax}(\text{Linear}(h_t^q))$. By Eq. (5), LLaMA-VQA has to generate the question which derives the answer from the video, leveraging its prior knowledge of temporal and causal reasoning.

**QA → V.** Another flipped task is video prediction given a question and an answer. It is formulated as:

$$\mathcal{L}_{\text{qav}} = - \log P(\mathbf{v}|\mathbf{q}, \mathbf{a})$$
$$= - \sum_{t=0}^{N_v-1} \log P(v_{t+1}|\mathbf{q}, \mathbf{a}, v_{\leq t}). \quad (6)$$

In contrast to the text generation loss in Eq. (2) and Eq. (5), which selects a token among the fixed vocabulary set (discrete space), it is too challenging to generate a video. So we instead adopt InfoNCE (Oord et al., 2018) to maximize the mutual information between the input frame feature $v_{t+1} \in \mathbb{R}^D$ and the output feature ($h_t^v \in \mathbb{R}^D$) of LLaMA-VQA. Then, the likelihood in Eq. (6) is calculated as:

$$P(v_{t+1}|\mathbf{q}, \mathbf{a}, v_{\leq t}) = \frac{\exp(v_{t+1}^\top h_t^v)}{\sum_{i=1}^{N_v} \exp(v_i^\top h_t^v)}, \quad (7)$$

where $h_0^v$ is the token representation right before the start of visual tokens. This encourages the model to predict the order of video frames given preceding frames, *i.e.*, next frame prediction, by analyzing the question and answer with LLMs' prior knowledge. This formulation enables video prediction via a unified text-generation-based QA model with minimum modification.

We combine all three objectives, which are LLMs' language generation losses and its variant. Finally, we train LLaMA-VQA with the following loss:

$$\mathcal{L}_{\text{Flipped-VQA}} = \mathcal{L}_{\text{vqa}} + \mathcal{L}_{\text{vaq}} + \mathcal{L}_{\text{qav}}. \quad (8)$$

We accumulate gradients of three different objectives and then update the learnable parameters.
*Remarks.* We observe that the objectives of the primary task and auxiliary tasks can be interpreted as learning *posterior* and *likelihood*, respectively. For instance, by the Bayes rule, we have

$P(\mathbf{a}|\mathbf{v}, \mathbf{q}) \propto P(\mathbf{q}|\mathbf{v}, \mathbf{a})P(\mathbf{a}|\mathbf{v})$. Hence, learning likelihood $P(\mathbf{q}|\mathbf{v}, \mathbf{a})$ via the question generation given a video and an answer benefits the primary task of predicting the answer given a video and a question, which is the posterior probability $P(\mathbf{a}|\mathbf{v}, \mathbf{q})$. Similarly, the same argument holds for video prediction; $P(\mathbf{a}|\mathbf{v}, \mathbf{q}) \propto P(\mathbf{v}|\mathbf{q}, \mathbf{a})$. These relationships explain why training a VQA model with flipped tasks boosts the performance of the target task. More detailed discussion is provided in Sec. E.

## 4 Experiments

We verify the effectiveness of our framework to leverage the powerful prior knowledge induced by an LLM. For a thorough analysis, our framework is applied to various LLMs: LLaMA (7B, 13B, and 33B), OPT (125M $\sim$ 6.7B), GPT-J (6B). We conduct experiments and analyses to answer the following research questions:

**Q1.** Do LLMs possess the knowledge of temporal and causal reasoning?
**Q2.** Is Flipped-VQA effective for dealing with challenging VideoQA?
**Q3.** How does Flipped-VQA alleviate linguistic bias?

**Datasets.** We experiment on five multiple-choice VideoQA benchmark datasets (NExT-QA, STAR, DramaQA, VLEP, and TVQA) which require challenging temporal and causal reasoning abilities. Further experimental settings and implementation details are provided in Sec. A and Sec. B.

### 4.1 Temporal and causal reasoning of LLMs

We investigate LLMs' strong prior of temporal and causal reasoning to answer **Q1** by comparing our framework with both LLMs-based and non-LLMs-based models for VideoQA.

**Comparison of various sizes of LLMs.** We first conduct the experiment on various LLMs sizes to verify the effectiveness of LLMs' temporal and causal reasoning ability on VideoQA in Fig. 3. Note that Flipped-VQA is not applied in this experiment to show that LLMs already possess strong reasoning ability themselves. In Fig. 3a, we evaluate various sizes of LLMs trained with entire $\langle$V, Q, A$\rangle$ triplets and the result shows that the performances on causal and temporal questions are dramatically improved as the model size increases. On the other hand, the performance gain of descriptive questions is relatively smaller than causal and temporal ques-

| Models | Language Model | # trainable params | NExT-QA | | | | STAR | | | | | DramaQA | VLEP | TVQA |
|---|---|---|---|---|---|---|---|---|---|---|---|---|---|---|
| | | | Cau. | Tem. | Des. | Tot. | Int. | Seq. | Pre. | Fea. | Tot. | Tot. | Tot. | Tot. |
| HGA (Jiang and Han, 2020) | GRU | - | 46.8 | 52.1 | 59.3 | 50.4 | - | - | - | - | - | - | - | - |
| FrozenBiLM (Yang et al., 2022) | DeBERTa | 30M | - | - | - | - | - | - | - | - | - | - | - | 82.0 |
| MERLOT (Zellers et al., 2021) | RoBERTa | 223M | - | - | - | - | - | - | - | - | - | 81.4 | 68.4 | 78.7 |
| HCRN (Le et al., 2021) | LSTM | 44M | 45.9 | 49.3 | 53.7 | 48.2 | - | - | - | - | - | - | - | - |
| SPCRL (Kim et al., 2021) | BERT | - | - | - | - | - | - | - | - | - | - | 81.0 | - | 76.2 |
| VGT (Xiao et al., 2022) | BERT | - | 53.4 | 56.4 | 69.5 | 56.9 | - | - | - | - | - | - | - | - |
| AIO (Wang et al., 2023a) | - | 110M | 48.0 | 48.6 | 63.2 | 50.6 | 47.5 | 50.8 | 47.8 | 44.1 | 47.5 | - | - | - |
| VidL (Cheng et al., 2023) | BERT | 25M | - | - | - | - | - | - | - | - | - | - | - | 79.0 |
| ATP (Buch et al., 2022) | CLIP text encoder | - | 53.1 | 50.2 | 66.8 | 54.3 | 50.6 | 52.9 | 49.4 | 40.6 | 48.4 | - | - | - |
| MIST (Gao et al., 2023) | - | - | 54.6 | 56.6 | 66.9 | 57.2 | 55.6 | 54.2 | 54.2 | 44.5 | 53.9 | - | - | - |
| HiTeA (Ye et al., 2022) | BERT | - | 62.4 | 58.3 | 75.6 | 63.1 | - | - | - | - | - | - | - | - |
| InternVideo (Wang et al., 2022) | CLIP text encoder | 1.3B | 62.5 | 58.5 | 75.8 | 63.2 | 62.7 | 65.6 | 54.9 | 51.9 | 58.7 | - | 63.9 | 57.2 |
| **LLaMA-VQA (Ours)** | LLaMA | **4.5M** | **72.7** | **69.2** | **75.8** | **72.0** | **66.2** | **67.9** | **57.2** | **52.7** | **65.4** | **84.1** | **71.0** | **82.2** |

Table 2: **Comparison on five challenging VideoQA benchmarks with non-LLMs-based models.** NExT-QA involves causal, temporal, and descriptive question types. STAR contains four question types: interaction, sequence, prediction, and feasibility. Total accuracy is highlighted in grey.

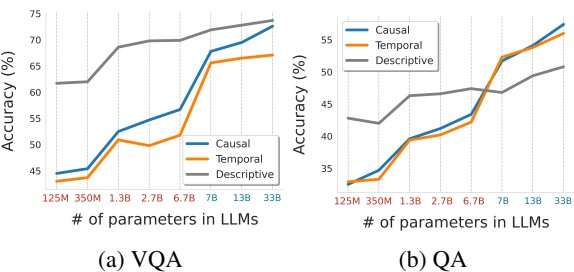

(a) VQA  (b) QA

Figure 3: **Performances of LLMs on three question types of NExT-QA.** Performances of various sizes of OPT (125M ∼ 6.7B) and LLaMA (7B ∼ 33B) are reported. A VideoQA approach with a larger language model achieves a better performance in both VQA and QA settings. Surprisingly, the QA approach with LLaMA (33B) outperforms VQA models with OPT (125M ∼ 6.7B) in temporal and causal reasoning.

| Models | LLMs | # total params | # trainable params | NExT-QA | | | |
|---|---|---|---|---|---|---|---|
| | | | | Cau. | Tem. | Des. | Tot. |
| BLIP-2 (Li et al., 2023a) | FlanT5 | 12.1B | 188M | 70.1 | 65.2 | 80.1 | 70.1 |
| SeViLA (Yu et al., 2023) | FlanT5 | 12.1B | 188M | 74.2 | 69.4 | **81.3** | 73.8 |
| **LLaMA-VQA** | LLaMA | 7B | **4.5M** | 72.7 | 69.2 | 75.8 | 72.0 |
| | LLaMA | 13B | 6M | 75.3 | 71.7 | 75.9 | 74.2 |
| | LLaMA | 33B | 9.2M | **76.2** | **72.6** | 78.8 | **75.5** |

Table 3: **Comparison with LLM-based models.**

tions. The performance gap between descriptive and causal questions has decreased from 17.2% on 125M to 1.1% on 33B.

Also, to verify the LLMs' prior knowledge of temporal and causal reasoning, we evaluate LLMs trained with only ⟨Q, A⟩ pairs by forcing the model to solely rely on the question. In Fig. 3b, with only linguistic information (*i.e.*, question), the performance of causal questions is lower than descriptive questions on 125M, but it significantly improves as the model size increases and outperforms the descriptive question accuracy on 33B by a margin of 6.6%. Without visual content, this model already outperforms MIST (Gao et al., 2023) in Tab. 2, a non-LLMs-based model, in terms of causal and temporal question types. These results suggest that larger LLMs possess more powerful prior of causal and temporal reasoning obtained during pretraining, and such prior plays a significant role in exploiting linguistic shortcuts for complex VideoQA.

**Comparison with non-LLMs-based models.** In Tab. 2, we then show the results of LLaMA-VQA in comparison to non-LLMs-based models on five challenging VideoQA benchmark datasets. Our LLaMA-VQA outperforms all the baselines across various datasets by a significant margin, especially on causal and temporal questions. For example, in NExT-QA, the performance gain in descriptive questions type is marginal compared to InternVideo, but it yields more than 10% improvements in both temporal and causal questions. Also, in STAR, LLaMA-VQA surpasses MIST on all question types resulting in an 11.5% increase in total accuracy. Particularly, the performance on sequence questions, which ask for temporal reasoning about consecutive actions, is improved by a large margin of 13.7%. These results highlight that LLMs-based LLaMA-VQA achieves remarkable capability, especially on temporal and causal reasoning questions compared to non-LLMs-based models, by mainly leveraging its pretrained prior (introducing only 4.5M learnable parameters).

**Comparison with LLMs-based models.** We also explore larger LLaMA-VQA (∼ 33B) and compare them with LLMs-based models on NExT-QA in Tab. 3. Our LLaMA-VQA 33B outperforms BLIP-2 and SeViLA in terms of the total accuracy only with 9.2M trainable parameters. Specifically, the performance gain on causal and temporal questions

| LLMs | Sizes | Epochs | Objectives $\mathcal{L}_{vqa}$ | $\mathcal{L}_{vaq}$ | $\mathcal{L}_{qav}$ | NExT-QA | STAR | DramaQA |
|---|---|---|---|---|---|---|---|---|
| OPT | 6.7B | 15 | ✔ | | | 57.0 | 57.7 | 73.0 |
| | | 5 | ✔ | | | 57.2 | 56.6 | 73.2 |
| | | 5 | ✔ | ✔ | | 60.9 | 60.0 | 76.9 |
| | | 5 | ✔ | ✔ | ✔ | **62.3** | **63.3** | **78.7** |
| GPT-J | 6B | 15 | ✔ | | | 62.8 | 59.3 | 80.7 |
| | | 5 | ✔ | | | 62.6 | 59.3 | 80.1 |
| | | 5 | ✔ | ✔ | | 64.2 | 60.1 | 81.1 |
| | | 5 | ✔ | ✔ | ✔ | **67.1** | **63.7** | **82.7** |
| LLaMA | 7B | 15 | ✔ | | | 67.1 | 60.6 | 82.4 |
| | | 5 | ✔ | | | 68.7 | 60.9 | 82.6 |
| | | 5 | ✔ | ✔ | | 71.2 | 6.1 | 83.3 |
| | | 5 | ✔ | ✔ | ✔ | **72.0** | **65.4** | **84.1** |

Table 4: **Comparison of various LLMs and objectives.** Total accuracy is reported.

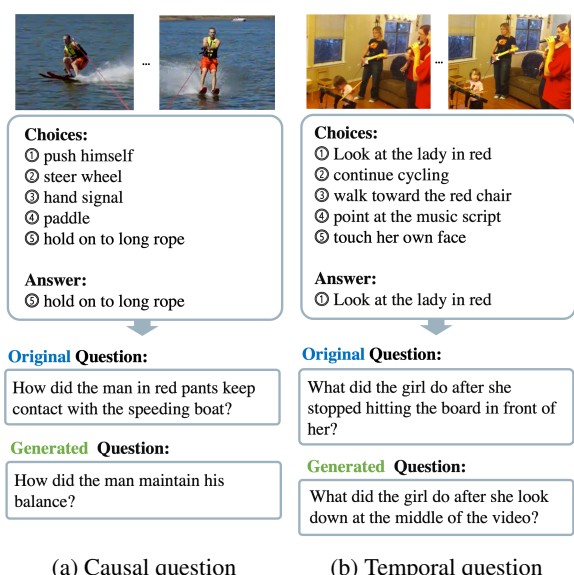

(a) Causal question      (b) Temporal question

Figure 4: **Examples of question generation.**

is 2% and 3.2% compared to SeViLA. On the other hand, the accuracy of LLaMA-VQA on descriptive questions is lower than baselines since they were further pretrained with large-scale image-caption pair datasets, which boosts the descriptiveness capability of visual content. Finally, as the model size of LLaMA-VQA increases (7B → 33B), the performance on causal and temporal questions is increased by 3.5% and 3.4%, respectively implying that larger LLMs have more powerful temporal and causal reasoning capabilities.

## 4.2 Flipped-VQA on challenging VideoQA

We here discuss **Q2** by analyzing the effectiveness of Flipped-VQA on challenging VideoQA.

**Ablation studies of Flipped-VQA.** Tab. 4 shows the ablation studies of Flipped-VQA on various LLMs (OPT, GPT-J, and LLaMA). Compared to the baseline LLMs with $\mathcal{L}_{vqa}$, introducing a question generation objective $\mathcal{L}_{vaq}$ improves the performances by 3.7%, 1.6%, and 2.5% in NExT-QA on OPT, GPT-J, and LLaMA, respectively. This result demonstrates that generating intricate questions of NExT-QA given a video-answer pair encourages LLMs to leverage their temporal and causal reasoning knowledge to predict the answer. In addition, further improvement is observed by adapting video predicting objective $\mathcal{L}_{qav}$ that helps to understand the order of visual contents based on the question and answer. The accuracy of GPT-J is increased by a margin of 2.9%, 3.6%, and 1.6% respectively on NExT-QA, STAR, and DramaQA. Overall, each component of Flipped-VQA improves the performance across various LLMs, implying that Flipped-VQA is an effective training objective for LLMs to deal with challenging VideoQA by leveraging their pretrained knowledge.

Unlike solely using $\mathcal{L}_{vqa}$ to perform the main task, Flipped-VQA accumulates gradients from three different objectives. So we conduct an additional experiment by increasing the gradient accumulation steps three times more than the baseline, *i.e.*, we additionally train $\mathcal{L}_{vqa}$ for 15 epochs while the others are trained for 5 epochs. In Tab. 4, the performance of $\mathcal{L}_{vqa}$ with 15 epochs is on par with the one with 5 epochs across various LLMs and datasets. This suggests that the performance gain of Flipped-VQA does not come from the increased gradient accumulation or training schedule, but comes from the capability of LLMs' prior knowledge exploited by $\mathcal{L}_{vaq}$ and $\mathcal{L}_{qav}$.

**Qualitative results of generated questions.** Fig. 4 illustrates examples of questions generated by LLaMA-VQA conditioned on the video and answer with the objective of $\mathcal{L}_{vaq}$, in the NExT-QA validation set. We observe that the majority of ⟨V, Q, A⟩ triplets with generated questions are plausible enough to answer, while the generated questions depict different aspects from the video than the originals. For instance of the causal question in Fig. 4a, LLaMA-VQA combines the visual content, a man waterskiing, with the answer "hold on to long rope" and expresses as the man is "maintain[ing] balance". Note that the idea of "keep[ing] contact" in the original question aligns with the idea of "maintain[ing] his balance", so there is no difficulty for the model to answer the generated question based on the given video. Hence it shows how LLMs are using their pretrained knowledge to generate the question appropriate to the given

| | Exploiting linguistic shortcut | | Mitigating linguistic bias | |
|---|---|---|---|---|
| | $P\left(\hat{Y}_{A|V,Q}=Y \left| \hat{Y}_{A|Q}=Y\right.\right)$ | | $P\left(\hat{Y}_{A|V,Q}=Y \left| \hat{Y}_{A|Q}\neq Y\right.\right)$ | |
| **Flipped-VQA** | ✘ | ✔ | ✘ | ✔ |
| **Ratio** | 82.7% | 87.3% | 50.7% | 53.8% |

Table 5: **Ratio of the number of samples.** $\hat{Y}_{A|Q}$ denotes the prediction of the model trained with only $\langle$Q, A$\rangle$. $\hat{Y}_{A|Q,V}$ stands for the prediction of LLaMA-VQA either trained with or without Flipped-VQA. $Y$ is a ground truth.

video and answer.

More interestingly, LLaMA-VQA is capable of understanding intricate interactions present in the video. For the temporal question in Fig. 4b, unlike the original question that asks for the action after the girl "stop[s] hitting the board", the generated question asks for the action after "look[ing] down". This reveals that LLaMA-VQA understands the interaction between objects and the sequential events in the video, and thus it can generate temporal questions adequate to the answer. These results suggest that LLaMA-VQA successfully understands the complex relationship between the video, question, and answer with Flipped-VQA by leveraging its prior knowledge of temporal and causal reasoning.

### 4.3 Flipped-VQA for mitigating linguistic bias

We observed that Flipped-VQA is crucial to address linguistic bias. So we finally analyze how Flipped-VQA alleviates linguistic bias (**Q3**) by providing detailed quantitative and qualitative analysis.

**Quantitative results of bias mitigation.** Although LLMs' strong prior of temporal and causal reasoning is beneficial to exploit linguistic shortcuts for challenging questions, this sometimes leads to suboptimal results by forcing the model to overly rely on the *text* question while ignoring the *visual* content. This problem, linguistic bias, is commonly observed in visual question answering (Niu et al., 2021; Ramakrishnan et al., 2018; Cadene et al., 2019). Our extensive analyses show that Flipped-VQA mitigates the *linguistic bias* while effectively leveraging *linguistic shortcut*. We first analyze how effectively LLaMA-VQA exploits linguistic shortcuts. Specifically, Tab. 5 shows that Flipped-VQA improves the accuracy of LLaMA-VQA on NExT-QA by 4.6% when the linguistic prior is correct. It is measured by the following equation:

$$P\left(\hat{Y}_{A|V,Q}=Y \left| \hat{Y}_{A|Q}=Y\right.\right). \quad (9)$$

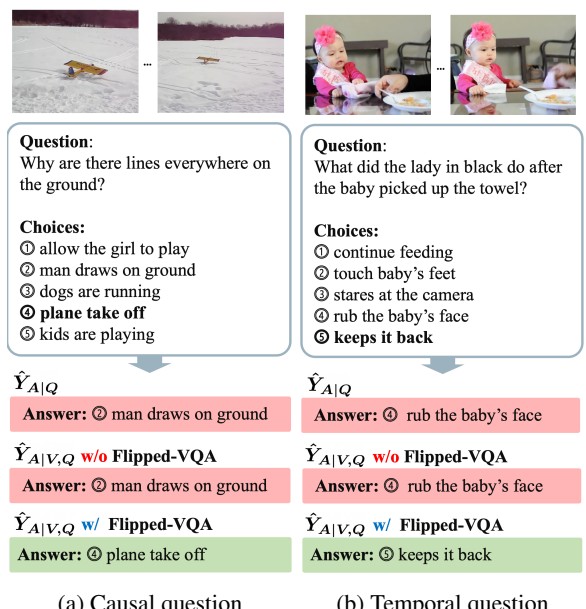

(a) Causal question      (b) Temporal question

Figure 5: **Examples of alleviation on linguistic bias.**

Here, the correctness of linguistic prior is defined as the accuracy of the QA model that predicts answers solely based on the language, *i.e.*, $P(\hat{Y}_{A|Q}=Y)$. Secondly, we analyze how effectively LLaMA-VQA mitigates linguistic bias by the following metric:

$$P\left(\hat{Y}_{A|V,Q}=Y \left| \hat{Y}_{A|Q}\neq Y\right.\right). \quad (10)$$

This measures the accuracy of the VQA model when the linguistic prior is *wrong*, *i.e.*, $\hat{Y}_{A|Q}\neq Y$. Tab. 5 shows that Flipped-VQA improves the accuracy on the samples with linguistic bias (inaccurate linguistic priors) by 3.1%. Our in-depth analysis of attention and embeddings also shows that Flipped-VQA encourages LLMs to leverage more visual content and better align the visual embedding space with LLMs' text embedding space, see Sec. F for details.

**Qualitative results of bias mitigation.** We here further analyze the effectiveness of Flipped-VQA in mitigating linguistic bias with qualitative results. Given incorrect linguistic prior, *i.e.*, $\hat{Y}_{A|Q}\neq Y$, in other words, when the prediction by the language-only model is wrong, our model trained with Flipped-VQA better rejects the wrong linguistic bias than the one trained without Flipped-VQA. For example in Fig. 5a the language-only model outputs "man draws on ground" for the causal question "Why are there lines everywhere on the ground?". LLaMA-VQA trained without Flipped-VQA fails to reject the wrong linguistic prior and chooses the

plausible-sounding answer based on the common knowledge in the pretrained language model without referring to visual content. This can be viewed as the hallucination and ungrounded guess problem observed in Alayrac et al. (2022). On the other hand, LLaMA-VQA trained with Flipped-VQA refers to the visual content that depicts a plane leaving traces on the snow as it is taking off and successfully predicts the actual cause "plane take off". The enhanced attention between the answer and visual tokens supports that Flipped-VQA encourages the model to refer to visual content, see Sec. F for more details.

Similarly, LLMs' temporal prior occasionally disrupts identifying the action followed by an event. For the temporal question in Fig. 5b, the act of wiping off follows the act of "pick[ing] up the towel" in general. Hence, the language-only model $\hat{Y}_{A|Q}$ and LLaMA-VQA $\hat{Y}_{A|V,Q}$ without Flipped-VQA predict "rub the baby's face." In contrast, the proposed method with Flipped-VQA accurately predicts "keeps it back" by referring to the video. These results demonstrate that Flipped-VQA encourages the answers grounded on visual information and mitigates linguistic bias.

## 5 Conclusion

In this paper, we investigate the large language models' (LLMs') temporal and causal reasoning abilities on the challenging multi-modal Video Question Answering (VideoQA) task. We observe that the larger LLMs possess more powerful prior knowledge of temporal and causal reasoning in addressing complex VideoQA. Moreover, we propose a novel framework, Flipped-VQA, that effectively leverages the LLMs' knowledge of temporal and causal reasoning on understanding the complex ⟨V, Q, A⟩ triplet by introducing three generative objectives: $\mathcal{L}_{\text{vqa}}$, $\mathcal{L}_{\text{vaq}}$, and $\mathcal{L}_{\text{qav}}$. Our in-depth analyses show that Flipped-VQA not only enhances the exploitation of linguistic shortcuts but also mitigates linguistic bias that causes hallucination and ungrounded guess problems.

**Acknowledgments.** This work was partly supported by ICT Creative Consilience program (IITP-2023-2020-0-01819) supervised by the IITP, the National Research Foundation of Korea (NRF) grant funded by the Korea government (MSIT) (NRF-2023R1A2C2005373), and KakaoBrain corporation.

## Limitations

We propose a Flipped-VQA which can be widely adaptable to decoder-only LLMs by improving their performances on challenging VideoQA. Flipped-VQA effectively leverages LLMs' prior knowledge of temporal and causal reasoning with generative objectives of next token prediction. Extending it to encoder-decoder LLMs with an objective other than the next token prediction can be interesting. Also, although the number of trainable parameters of LLaMA-VQA is only 4.5M ∼ 9.2M, the total number of parameters is inherently large (7B ∼ 33B), which mainly comes from backbone LLMs, LLaMA. This leads to massive memory usage during training/fine-tuning and inference.

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

# Appendix

## A  Dataset details

**NExT-QA** (Xiao et al., 2021) consists of three types of questions. Causal questions ask for the intentions of earlier actions or reasons for succeeding ones. Temporal questions determine the relationships between actions that are solely based on the sequence of occurrence (*e.g. what ... do after/before/while ...*). Descriptive questions focus on visible contents such as places, and objects/attributes. About 5K videos of average 44s and 48K QA pairs with five answer candidates are given. Further examples of each question type are provided in Sec. D.

**DramaQA** (Choi et al., 2021) features video story understanding with hierarchical difficulty levels. The level is determined by the required length of the clip (shot or scene) and the number of logical reasoning steps to answer the question. The dataset contains 24K video clips and 18K QA pairs with five answer candidates. Average video lengths are 3.7s for the shot and 91.3s for the scene.

**STAR** (Wu et al., 2021) is designed for situational reasoning with questions that tackle interaction, sequence, prediction, and feasibility of events. There exist 60K QA pairs with four answer candidates and 22K video clips.

**VLEP** (Lei et al., 2020) uses TV shows and YouTube Vlogs with an average of 6.1s that contain rich physical interactions and dialogues between people. The challenge is to determine which of two future events is likely to occur in the given video (with dialogue). It is comprised of 29K QA pairs with 10K video clips.

**TVQA** (Lei et al., 2018) is built on long video clips (60-90s) of six different TV shows with various social interactions and activities. It provides dialogues for each video with 153K QA pairs and 22K video clips.

## B  Implementation details

**Training details.** LLaMA-VQA is trained for five epochs on all datasets with a batch size of 32. LLaMA-VQA 7B and 13B are trained with $8 \times$ A6000 GPUs and LLaMA-VQA 33B is trained with $8 \times$ A100 GPUs. AdamW optimizer (Loshchilov and Hutter, 2017) is used with $\beta = (0.9, 0.95)$. We search learning rate and weight decay in [0.05, 0.1] and [0.15, 0.25], respectively. Following LLaMA-Adapter, for each layer of LLMs, 10 adapter tokens are used, *i.e.* $N_p = 10$. The number of video frames $N_v$ is set to 10. Each frame is resized by $224 \times 224$ and fed into CLIP VIT-L/14 (Radford et al., 2021) to extract frame features. The total sequence length of the concatenated visual, question, and answer tokens, $N_v + N_q + N_a$, is 128, 128, 384, 256, and 512 for NExT-QA, STAR, DramaQA, VLEP, and TVQA respectively. Each dataset optionally provides dialogues. We append dialogues as prefix tokens for

```
[SOS] Video: <v₁><v₂>···<v_{N_v}>
Question: <question>
Choices:
(A) <option 1>
(B) <option 2>
(C) <option 3>
(D) <option 4>
(E) <option 5>
Answer: The answer is <answer> [EOS]
```

Table 6: **Input Prompt of VQ → A.**

```
[SOS] Video: <v₁><v₂>···<v_{N_v}>
Choices:
(A) <option 1>
(B) <option 2>
(C) <option 3>
(D) <option 4>
(E) <option 5>
Answer: The answer is <answer> [EOS]
Question: <question> [EOS]
```

Table 7: **Input Prompt of VA → Q.**

```
[SOS] Question: <question>
Choices:
(A) <option 1>
(B) <option 2>
(C) <option 3>
(D) <option 4>
(E) <option 5>
Answer: The answer is <answer>
Video: <v₁><v₂>···<v_{N_v}>
```

Table 8: **Input Prompt of QA → V.**

VLEP and TVQA. $\mathcal{L}_{\text{vaq}}$ is not applied in VLEP since questions of all samples in VLEP are consistent to "Which event is more likely to happen right after?".

**Prompt details.** The general input prompt of LLaMA-VQA is provided in Tab. 1. Also, Tab. 6, Tab. 7, and Tab. 8 provides detailed input prompt of each task in Flipped-VQA, respectively. In those tables, non-prefix tokens, which the model needs to generate, are highlighted in red and the rest are prefix tokens.

## C  LLaMA-Adapter

LLaMA-Adapter (Zhang et al., 2023c) adopts a set of learnable adapter tokens $\mathbf{p} = [p_1, \ldots, p_{N_p}] \in \mathbb{R}^{N_p \times D}$ to efficiently fine-tune LLaMA (Touvron et al., 2023), where $N_p$ is the number of adapter tokens and $D$ is a feature dimension. The adapter tokens are then concatenated as prefix tokens for the key and value of each self-attention layer, formulated as:

$$
\begin{aligned}
Q &= \text{Linear}_q([\mathbf{v}, \mathbf{q}, \mathbf{a}]) \in \mathbb{R}^N, \\
K &= \text{Linear}_k([\mathbf{p}, \mathbf{v}, \mathbf{q}, \mathbf{a}]) \in \mathbb{R}^{N_p+N}, \quad (11) \\
V &= \text{Linear}_v([\mathbf{p}, \mathbf{v}, \mathbf{q}, \mathbf{a}]) \in \mathbb{R}^{N_p+N},
\end{aligned}
$$

where $N = N_v + N_q + N_a$. Note in our work, according to each objective of Flipped-VQA, the order of $\mathbf{v}$, $\mathbf{q}$, and $\mathbf{a}$ is permuted. Then, the scaled dot-product between $Q$ and $K$ is calculated as:

$$
S = QK^\top/\sqrt{D} \in \mathbb{R}^{N \times (N_p+N)}. \quad (12)
$$

$S$ in Eq. (12) can be divided into two groups as:

$$
S = [S^{N_p}, S^N]^\top, \quad (13)
$$

where $S^{N_p} \in \mathbb{R}^{N \times N_p}$ is the attention scores of adapter tokens $\mathbf{p}$ and $S^N \in \mathbb{R}^{N \times N}$ is for $\mathbf{v}$, $\mathbf{q}$, $\mathbf{a}$ tokens.

Moreover, to adjust the contribution of newly adopted tokens $\mathbf{p}$ at the beginning of training, LLaMA-Adapters introduce a zero-init attention gate $g$ as:

$$
S = [\text{Softmax}(S^{N_p}) \cdot g, \text{Softmax}(S^N)]^\top, \quad (14)
$$

where $g$ is initialized to zero. Eq. (14) preserves the LLMs' knowledge at the beginning of training and gradually increases the influence of adapter tokens $\mathbf{p}$. Finally, the output of LLaMA-Adapter is as follows:

$$
H = \text{Linear}(SV) \in \mathbb{R}^{N \times D}. \quad (15)
$$

## D  NExT-QA

In NExT-QA, causal and temporal questions account for 48% and 29% respectively of the dataset. Specifically, there exist three types of questions: Causal, Temporal, and Descriptive. In general, questions and answers for temporal and causal types are longer than descriptive ones.

Causal questions seek for event A which happens in advance of event B and is also responsible for B's occurrence in the video. These questions are broken down into asking "how" such an event

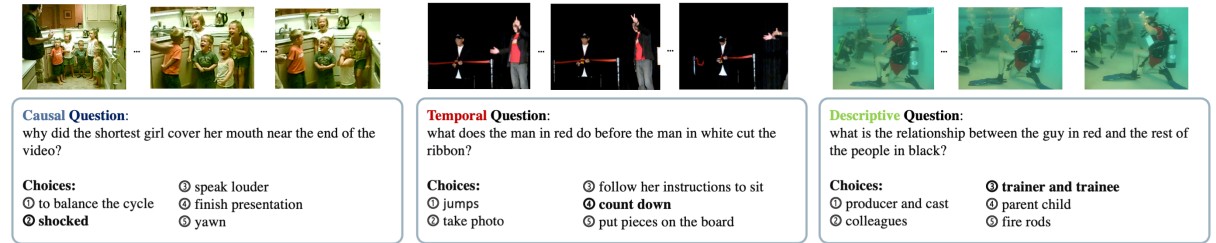

**Causal Question:**
why did the shortest girl cover her mouth near the end of the video?

**Choices:**
① to balance the cycle
② **shocked**
③ speak louder
④ finish presentation
⑤ yawn

**Temporal Question:**
what does the man in red do before the man in white cut the ribbon?

**Choices:**
① jumps
② take photo
③ **count down**
④ follow her instructions to sit
⑤ put pieces on the board

**Descriptive Question:**
what is the relationship between the guy in red and the rest of the people in black?

**Choices:**
① producer and cast
② colleagues
③ **trainer and trainee**
④ parent child
⑤ fire rods

Figure 6: **Examples of NExT-QA.**

occurred or "why" the object acts in a certain way. For instance, Fig. 6 (left) asks "Why did the shortest girl cover her mouth near the end of the video?".

Temporal questions are closely related to causality but require reasoning solely based on the sequence of occurrence (present, previous, or next actions) and further ask to focus on interactions of multiple objects. For example, a question regarding the previous action asks "What does the man in red do *before* the man in white cut the ribbon?" in Fig. 6 (middle).

Lastly, descriptive questions tend to ask about the video in general (*i.e.*, the places, objects/attributes, and main actions/events). For instance, Fig. 6 (right) asks "What is the relationship between the guy in red and the rest of the people in black?".

## E    Discussion on Flipped-VQA

The primary task of VideoQA, predicting the answer given the video and question, can be rewritten as:

$$P(\mathbf{a}|\mathbf{v},\mathbf{q}) = \frac{P(\mathbf{v}|\mathbf{a},\mathbf{q})P(\mathbf{a}|\mathbf{q})}{P(\mathbf{v}|\mathbf{q})} \propto P(\mathbf{v}|\mathbf{a},\mathbf{q}). \tag{16}$$

In Eq. (16), we observe that the auxiliary task of predicting visual tokens $\mathbf{v}$ given $\mathbf{q}$ and $\mathbf{a}$ can be viewed as maximum *likelihood* estimation (*i.e.*, $P(\mathbf{v}|\mathbf{a},\mathbf{q})$) and the primary task as the maximum a *posterior* (MAP) estimation (*i.e.*, $P(\mathbf{a}|\mathbf{v},\mathbf{q})$), respectively.

Similarly, the auxiliary task of predicting the question token $\mathbf{q}$ given $\mathbf{a}$ and $\mathbf{v}$ can be correlated with the primary task as:

$$P(\mathbf{a}|\mathbf{v},\mathbf{q}) = \frac{P(\mathbf{q}|\mathbf{a},\mathbf{v})P(\mathbf{a}|\mathbf{v})}{P(\mathbf{q}|\mathbf{v})} \propto P(\mathbf{q}|\mathbf{a},\mathbf{v}). \tag{17}$$

Therefore, by maximizing the likelihoods of auxiliary tasks, $\mathcal{L}_{\text{Flipped-VQA}}$ in Eq. (8) aims to strengthen the performance on the main task.

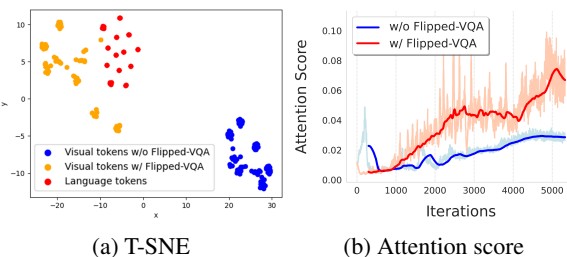

(a) T-SNE    (b) Attention score

Figure 7: **Visualization of T-SNE and attention score.** (a) Each input token embeddings of visual $\mathbf{v}$ and question $\mathbf{q}$ is visualized. (b) Attention score between answer query tokens and visual key tokens is visualized.

## F    Embedding space alignment of Flipped-VQA

To bridge the visual encoder embedding space with LLMs text embedding space, we adopt a simple linear projection layer $f$. We observe that without Flipped-VQA, $f$ is not trained effectively as the visual tokens $\mathbf{v}$ are just used as prefix tokens which are excluded from the generation target of LLMs. However, with Flipped-VQA which directly propagates the loss to the visual tokens (red arrows of $\mathcal{L}_{\text{qav}}$ in Fig. 2), $f$ is trained to align visual encoder embedding space with frozen LLMs text token embedding space. We visualize the embedding space of question tokens $\mathbf{q}$ and visual tokens $\mathbf{v}$ with and without Flipped-VQA in Fig. 7a, and show that the embedding space of visual tokens with Flipped-VQA (orange) is closer to LLMs' text embedding space (red) compared to the one without Flipped-VQA (blue).

Furthermore, in Fig. 7b, we plot the attention score between the answer query tokens and visual key tokens, *i.e.*, measuring how much visual tokens $\mathbf{v}$ affect answer tokens $\mathbf{a}$. As training proceeds, the attention score of both LLaMA-VQA with and without Flipped-VQA gradually increases, representing that the model leverages more visual content to answer the question. However, after the entire training iterations, the attention score with

| Models | # external visual-text data samples | WUPS |
|---|---|---|
| HGA (Jiang and Han, 2020) | 0 | 25.2 |
| KcGA (Jin et al., 2023) | 0 | 28.2 |
| Flamingo 0-shot (Alayrac et al., 2022) | 2.1B | 26.7 |
| Flamingo 32-shot (Alayrac et al., 2022) | 2.1B | 33.5 |
| **LLaMA-VQA** (Ours) | **0** | **34.3** |

Table 9: **Results of NExT-QA.**

| Models | # external visual-text data samples | Accuracy |
|---|---|---|
| JustAsk (Yang et al., 2021) | 69M | 38.9 |
| SiaSamRea (Yu et al., 2021) | 5.6M | 39.8 |
| MERLOT (Zellers et al., 2021) | 180M | 41.4 |
| FrozenBiLM (Yang et al., 2022) | 10M | 43.2 |
| Singularity (Lei et al., 2023) | 17M | 44.1 |
| FrozenBiLM+ (Ko et al., 2023) | 10M | 44.8 |
| UMT-L (Li et al., 2023c) | 25M | 47.9 |
| **LLaMA-VQA** (Ours) | **0** | **48.6** |

Table 10: **Results of ActivityNet-QA.**

Flipped-VQA (red) is three times larger than the one without Flipped-VQA (blue), indicating that Flipped-VQA plays a key role in transferring the visual representation into the LLMs embedding space and enhances the utilization of visual content on LLMs. These results demonstrate that Flipped-VQA encourages text-only trained LLMs to understand visual content by utilizing the strong representation power of a pretrained visual encoder, effectively aligning the visual embedding space with the text embedding space.

## G Further quantitative results

We conduct an additional experiment on two generation-based VideoQA benchmark datasets: NExT-QA open-form generation (Xiao et al., 2021) and ActivityNet-QA (Yu et al., 2019). WUPS and Accuracy are used for evaluation metrics in NExT-QA open-form generation and ActivityNet-QA, respectively. In Tab. 9, our LLaMA-VQA outperforms non-LLMs-based models HGA and KcGA by a large margin. Also, compared with LLMs-based Flamingo which is further trained with 2.1B external visual-text pairs, the performance gain is 0.8%. Furthermore, in Tab. 10, LLaMA-VQA also outperforms all the baselines although those use large-scale visual-text data for extra training in ActivityNet-QA.

## H Further qualitative results

**Question generation of Flipped-VQA.** We here show further qualitative examples of generated questions by $\mathcal{L}_{\text{vaq}}$ in Fig. 8. For the example in the middle of the first row, the generated question successfully describes the video, where the man moves across the muddy area, so the model can answer "jeep" based on this question. Also, LLaMA-VQA generates questions by referring to different timestamps of the video. In the right example of the first row, the original question asks the reason why the lioness bends its head in the middle of the video to lead to the answer "drink water". However, the lioness also touches the river to drink water at the beginning of the video, and the generated question asks the reason for touching the river to obtain the answer "drink water". This suggests that $\mathcal{L}_{\text{vaq}}$ of Flipped-VQA helps to understand the complex relationship between video, question, and answer with LLMs' prior knowledge.

**Bias alleviation of Flipped-VQA.** In addition to the examples in Fig. 5, we provide further qualitative results that Flipped-VQA enables the alleviation of linguistic bias. For the left example of the first row in Fig. 9, the model trained with question-answer pairs ($\hat{Y}_{A|Q}$) predicts "microphone" for the temporal question "What are the men holding on to when they speak?", based on the prior knowledge that people usually use the microphone when they are speaking. LLaMA-VQA trained without Flipped-VQA still fails to reject the wrong linguistic bias and predicts the same answer. On the other hand, LLaMA-VQA trained with Flipped-VQA accurately outputs the answer "glasses" by taking into account the video where the men are holding glasses, resulting in the reduction of hallucination problems with the mitigation of linguistic bias.

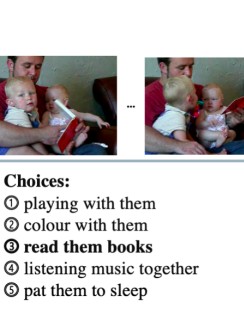

**Choices:**
① playing with them
② colour with them
③ **read them books**
④ listening music together
⑤ pat them to sleep

**Answer:**
③ read them books

**Original Question:**
Why does the man in red place the children on his lap?

**Generated Question:**
Why are the babies sitting on the man's lap?

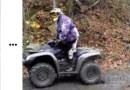 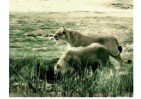

**Choices:**
① cycle up slope
② **jeep**
③ using the sled
④ handles on the machine
⑤ spread arms out

**Answer:**
② jeep

**Original Question:**
How does the man transport himself across?

**Generated Question:**
How did the people move across the muddy area?

**Choices:**
① playing
② distracted
③ pick up something
④ **drink water**
⑤ to make the dog jump

**Answer:**
④ drink water

**Original Question:**
Why did the lioness bend its head down in the middle?

**Generated Question:**
Why did the lioness touch the river in the beginning?

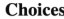

**Choices:**
① playing with ball
② **watch**
③ watching television
④ reading a book
⑤ sleeping

**Answer:**
② watch

**Original Question:**
What is the person in black behind the boy doing as the boy plays with the train?

**Generated Question:**
What is the adult doing?

**Choices:**
① share with the girl
② approach lady sitting there
③ **unwrap it**
④ playing with toy train
⑤ gesture something

**Answer:**
③ unwrap it

**Original Question:**
Why did the boy pick up one present from the group of them and move to the sofa?

**Generated Question:**
Why does the boy bend down and pick up the thing at the start of the video?

**Choices:**
① bite it
② sit there
③ ran away
④ **play with the hand**
⑤ play with cat

**Answer:**
④ play with the hand

**Original Question:**
What did the cat do after the human puts the hand on the floor?

**Generated Question:**
How did the person interact with the cat?

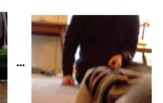

**Choices:**
① white cake
② cardboard box
③ letters
④ birthday gift
⑤ **food**

**Answer:**
⑤ food

**Original Question:**
What is on the table?

**Generated Question:**
What is in front of the girl?

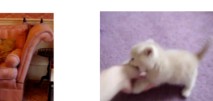

**Choices:**
① crawling
② grabbing on to the cot
③ sit up
④ **tear bit by bit**
⑤ sitting on table

**Answer:**
④ tear bit by bit

**Original Question:**
How does the kid open the present?

**Generated Question:**
How did the boy in red disassemble the wrapping paper from the toy at the start?

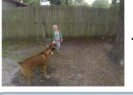

**Choices:**
① to catch ball
② **pick it up**
③ to not step on it
④ to bounce the balloon
⑤ kick balloon away

**Answer:**
② pick it up

**Original Question:**
Why does the boy run towards the balloon when it is on the ground?

**Generated Question:**
Why does the boy bend down towards the end of the video?

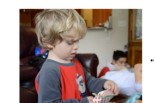 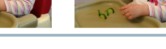 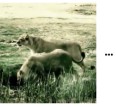 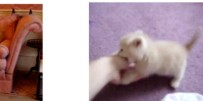 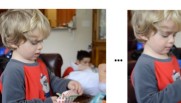

Figure 8: **Examples of question generation.**

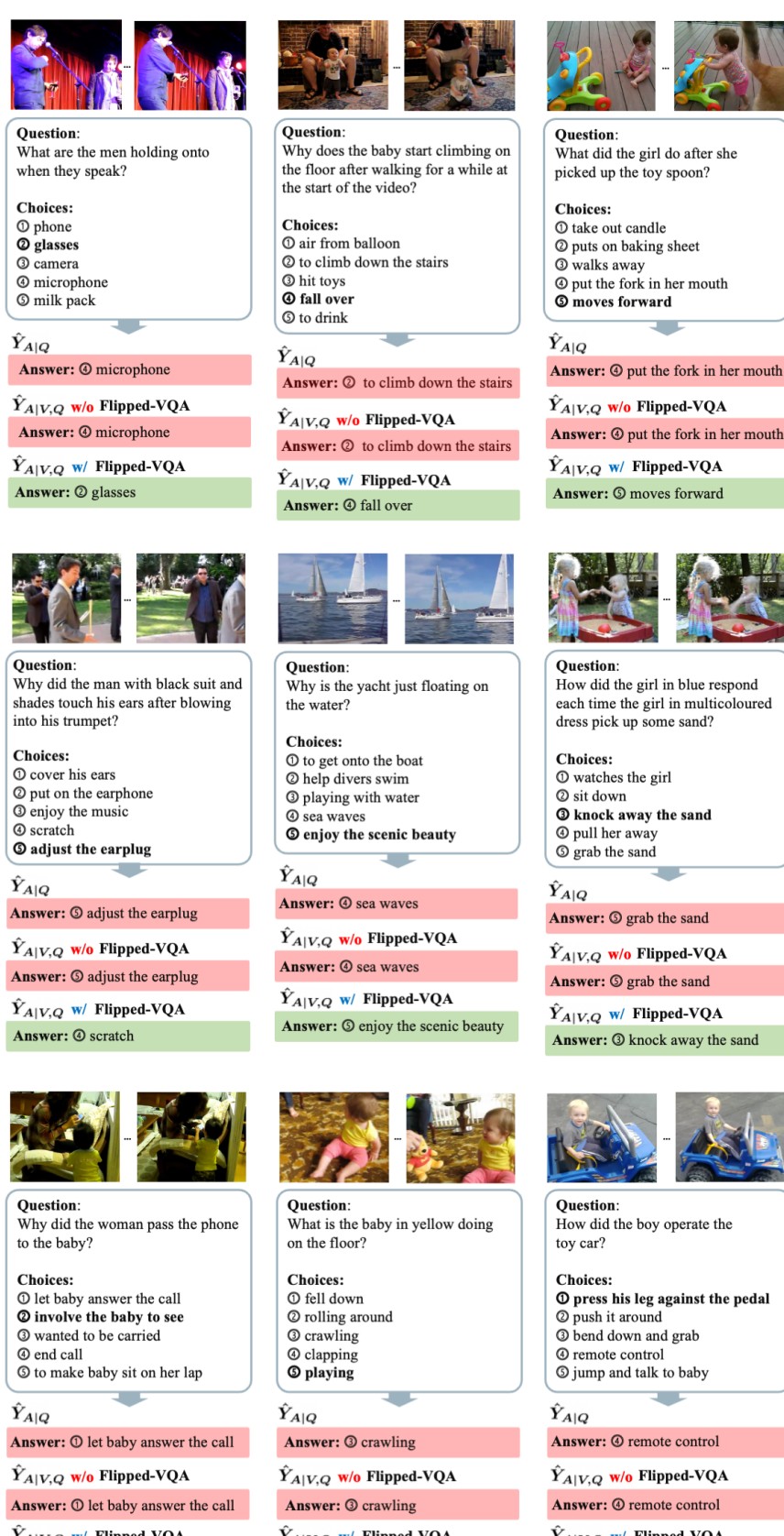

Figure 9: **Examples of alleviation on linguistic bias.**