# OpenReview forum: "Large Language Models are Temporal and Causal Reasoners for Video Question Answering"
_EMNLP/2023/Conference — EMNLP 2023 Main_

### Official Review · Reviewer_W2ex · 2023-07-31

**Paper Topic And Main Contributions:** 1. This paper proposes to utilize an …
**Soundness:** 4

**Excitement:**

4: Strong: This paper deepens the understanding of some phenomenon or lowers the barriers to an existing research direction.

**Reasons To Accept:**

1. Using LLMs for visual tasks is interesting and maybe an approach for more general artificial intelligence.
2. The proposed framework Flipped-VQA is simple but effective, as shown in Table 4, every part is beneficial to the final performance.
3. Extensive experiments have been conducted and the comparison between many existing methods have shown the superiority of their method.

**Reasons To Reject:**

1. There maybe other video prediction (QA->V) methods, e.g., a real video generation head. More exploration on this task is better.

**Reproducibility:**

4: Could mostly reproduce the results, but there may be some variation because of sample variance or minor variations in their interpretation of the protocol or method.

**Reviewer Confidence:**

3: Pretty sure, but there's a chance I missed something. Although I have a good feel for this area in general, I did not carefully check the paper's details, e.g., the math, experimental design, or novelty.

---

> ### Author Rebuttal · Authors · 2023-08-28
>
> We sincerely appreciate your strong acknowledgment of our paper. We will address your concerns and provide further implementation details in terms of reproducibility, hoping for more vigorous support for our paper.
>
> ---
>
> **Response 1: Exploration of other video prediction methods.**
>
> We explore two additional video prediction methods: video raw pixel generation and video feature generation. Both adopt L2 regression loss. In NExT-QA, Flipped-VQA with next frame prediction outperforms both raw pixel generation and feature generation, suggesting that the next frame prediction task is more beneficial for the temporal and causal understanding task.
> |  | Total |
> | --- | --- |
> | Raw pixel generation | 65.6 |
> | Feature generation | 71.7 |
> | **Next frame prediction (Ours)** | **72.4** |
>
> ---
>
> **Response 2: Further implementation details.**
>
> First, the general input prompt of LLaMA-VQA is provided in Tab. 1 of the main paper. Specifically, the table represents the prompt for VQ → A, where the prompt from `[SOS]` to `Answer:` serves as a prefix and the model needs to generate the tokens after `Answer:`. Similarly, for VA → Q, `Video: ...`, `Choices: ...`, and `Answer: ...` tokens serve as prefix tokens and onwards is for a generation. The same manner is applied to QA → V.
>
> Also, LLaMA-VQA 7B is composed of 32 heads and 32 layers and the feature dimension $D$ is 4096. The total number of vocabularies used is 32,000 from SentencePiece Tokenizer. For visual encodings, each frame is resized by 224 $\times$ 224 and fed into CLIP VIT-L/14 to extract frame features.
>
> The hyperparameters of our main experiments (Tab.2 of the main paper) are shown in the table below.
> |  | Learning Rate | # adapter tokens ($N_p$) | Batch Size | Weight Decay | Epoch | Warmup Epoch | # frames ($N_v$) | Max Sequence Length ($N_v + N_q + N_a$) |
> | --- | --- | --- | --- | --- | --- | --- | --- | --- |
> | NExT-QA | 0.0225 | 10 | 64 | 0.16 | 5 | 2 | 10 | 128 |
> | STAR | 0.0175 | 10 | 64 | 0.16 | 5 | 2 | 10 | 128 |
> | DramaQA | 0.0225 | 10 | 64 | 0.16 | 5 | 1 | 10 | 384 |
> | VLEP | 0.02 | 10 | 64 | 0.25 | 5 | 2 | 10 | 256 |
> | TQVA | 0.00875 | 10 | 32 | 0.02 | 5 | 2 | 10 | 512 |
>
> We will include further implementation details in our final version and release our code if accepted.

---

### Official Review · Reviewer_peaG · 2023-08-02

**Soundness:** 5

**Excitement:**

4: Strong: This paper deepens the understanding of some phenomenon or lowers the barriers to an existing research direction.

**Missing References:**

Some recent studies on Video-LLMs can be incorporated as related work:

[1] Video-LLaMA: An Instruction-tuned Audio-Visual Language Model for Video Understanding. Zhang et al.

[2] VideoChat: Chat-Centric Video Understanding. Li et al.

[3] M3IT: A Large-Scale Dataset towards Multi-Modal Multilingual Instruction Tuning. Li et al.

**Paper Topic And Main Contributions:**

This paper proposes a novel framework called Flipped-VQA, which trains a VideoQA model with three objectives: (1) predict the answer A based on video V and question Q, (2) predict question Q based on V and A, and (3) predict video V from Q and A. The authors show that adding objectives (2) and (3) improves performance on the primary VideoQA task while also reducing linguistic bias. The framework is implemented in LLaMA-VQA, which outperforms both LLMs-based and non-LLMs-based models on five VideoQA benchmarks.

**Reasons To Accept:**

1. The paper is well-written and easy to follow.
2. The Flipped-VQA idea is creative, expanding perspectives on utilizing VQA data. Interpreting the tasks as learning posterior and likelihood provides insight.
3. Experiments are solid and comprehensive, with comparisons to strong baselines including InternVideo (non-LLm-based) and BLIP-2 (LLM-based), and improvements over these baselines are significant. The authors also conduct extensive ablation study to verify the effectiveness of the two auxiliary tasks.

**Reasons To Reject:**

I don’t find significant flaws in this paper. There are some minor suggestions:

1. The VideoQA benchmarks in the paper are all choice-based. It would be better to choose some generation-based VideoQA datasets like ActivityNet-QA to increase the diversity.
2. I believe the Flipped-QA is a general framework for various generative VideoQA models.  However, the authors only apply this framework to LLM-based models. It would be better to further verify the effectiveness and  universality to non-LLM-based models like HiTeA and InternVideo.

**Reproducibility:**

3: Could reproduce the results with some difficulty. The settings of parameters are underspecified or subjectively determined; the training/evaluation data are not widely available.

**Reviewer Confidence:**

4: Quite sure. I tried to check the important points carefully. It's unlikely, though conceivable, that I missed something that should affect my ratings.

---

> ### Author Rebuttal · Authors · 2023-08-28
>
> We sincerely appreciate your strong acknowledgment of our paper. We will address your concerns and provide further implementation details in terms of reproducibility, hoping for more vigorous support for our paper.
>
> ---
>
> **Response 1: Experiments on generation-based VideoQA.**
>
> We conduct an additional experiment on two generation-based VideoQA benchmark datasets: NExT-QA open-form generation and ActivityNet-QA. WUPS and Accuracy are used for evaluation metrics in NExT-QA open-form generation and ActivityNet-QA, respectively. In the below table, our LLaMA-VQA outperforms non-LLMs-based models HGA and KcGA by a large margin. Also, compared with LLMs-based Flamingo which is further trained with 2.1B external visual-text pairs, the performance gain is 0.8%.
> |  | # external visual-text data samples | NExT-QA open-form generation (WUPS) |
> | --- | --- | --- |
> | HGA | 0 | 25.2 |
> | KcGA | 0 | 28.2 |
> | Flamingo 0-shot | 2.1B | 26.7 |
> | Flamingo 32-shot | 2.1B | 33.5 |
> | **LLaMA-VQA (Ours)** | **0** | **34.3** |
>
> Furthermore, in ActivityNet-QA, LLaMA-VQA also outperforms all the baselines although those use large-scale visual-text data for extra training.
> |  | # external visual-text data samples | ActivityNet-QA (Accuracy) |
> | --- | --- | --- |
> | JustAsk | 69M | 38.9 |
> | SiaSamRea | 5.6M | 39.8 |
> | MERLOT | 180M | 41.4 |
> | FrozenBiLM | 10M | 43.2 |
> | Singularity | 17M | 44.1 |
> | UMT-L | 25M | 47.9 |
> | **LLaMA-VQA (Ours)** | **0** | **48.6** |
>
> We compared our method with peer-reviewed works in the table aboves. But we will include discussions/comparisons with more recent/concurrent works, *e.g.*,  VALOR [1] and VAST [2], in the final version.
>
> [1] Chen et al., Valor: Vision-audio-language omni-perception pretraining model and dataset, 2023.
>
> [2] Chen et al., VAST: A Vision-Audio-Subtitle-Text Omni-Modality Foundation Model and Dataset, 2023.
>
> ---
>
> **Response 2: Adaptation of Flipped-VQA to non-LLMs-based models.**
>
> We apply Flipped-VQA to InternVideo, one of the recent strong non-LLMs-based baselines, to verify the broad applicability of our method. We modify the question generation and video prediction task as a contrastive learning task, which encourages matching VA and Q (or QA and V) pairs among batch samples, in order to apply Flipped-VQA to non-LLMs-based model. In NExT-QA, the performance gain is 0.8%.
> |  | Total |
> | --- | --- |
> | InternVideo | 63.2 |
> | **InternVideo + Flipped-VQA** | **64.0** |
>
> ---
>
> **Response 3: Missing references on Video-LLMs.**
>
> Thank you for the suggestions. Our work is studied concurrently with those works for Video-LLMs. We will include the mentioned works in our final version.
>
> ---
>
> **Response 4: Further implementation details.**
>
> First, the general input prompt of LLaMA-VQA is provided in Tab. 1 of the main paper. Specifically, the table represents the prompt for VQ → A, where the prompt from `[SOS]` to `Answer:` serves as a prefix and the model needs to generate the tokens after `Answer:`. Similarly, for VA → Q, `Video: ...`, `Choices: ...`, and `Answer: ...` tokens serve as prefix tokens and onwards is for a generation. The same manner is applied to QA → V.
>
> Also, LLaMA-VQA 7B is composed of 32 heads and 32 layers and the feature dimension $D$ is 4096. The total number of vocabularies used is 32,000 from SentencePiece Tokenizer. For visual encodings, each frame is resized by 224 $\times$ 224 and fed into CLIP VIT-L/14 to extract frame features.
>
> The hyperparameters of our main experiments (Tab.2 of the main paper) are shown in the table below.
> |  | Learning Rate | # adapter tokens ($N_p$) | Batch Size | Weight Decay | Epoch | Warmup Epoch | # frames ($N_v$) | Max Sequence Length ($N_v + N_q + N_a$) |
> | --- | --- | --- | --- | --- | --- | --- | --- | --- |
> | NExT-QA | 0.0225 | 10 | 64 | 0.16 | 5 | 2 | 10 | 128 |
> | STAR | 0.0175 | 10 | 64 | 0.16 | 5 | 2 | 10 | 128 |
> | DramaQA | 0.0225 | 10 | 64 | 0.16 | 5 | 1 | 10 | 384 |
> | VLEP | 0.02 | 10 | 64 | 0.25 | 5 | 2 | 10 | 256 |
> | TQVA | 0.00875 | 10 | 32 | 0.02 | 5 | 2 | 10 | 512 |
>
> We will include further implementation details in our final version and release our code if accepted.

---

### Official Review · Reviewer_iJms · 2023-08-05

**Typos Grammar Style And Presentation Improvements:** The paper is generally well written.
**Soundness:** 4

**Excitement:**

4: Strong: This paper deepens the understanding of some phenomenon or lowers the barriers to an existing research direction.

**Paper Topic And Main Contributions:**

The authors show that even though LLMs can exploit linguist shortcuts in VQA tasks, they can leverage the powerful LLMs with further tuning to reduce the linguistic bias and boost results. To achieve this, the authors propose to further train the LLM model with V-Q-A triples permuted and LLM frozen. Through extensive experiments, the authors show that the proposed Flipped-VQA achieves very good results on many benchmarks including NExT-QA, STAR, 332 DramaQA, VLEP, and TVQA.

**Reasons To Accept:**

1. The experiments are thorough and the results are convincing.
2. The authors decouple the reasoning ability of LLMs into temporal and causal and show good insights into reducing the linguist bias.

**Reasons To Reject:**

No particular reason to reject. The paper conveys good results and is well written.

**Reproducibility:**

3: Could reproduce the results with some difficulty. The settings of parameters are underspecified or subjectively determined; the training/evaluation data are not widely available.

**Reviewer Confidence:**

3: Pretty sure, but there's a chance I missed something. Although I have a good feel for this area in general, I did not carefully check the paper's details, e.g., the math, experimental design, or novelty.

---

> ### Author Rebuttal · Authors · 2023-08-28
>
> We sincerely appreciate your strong acknowledgment of our paper. We will provide further implementation details in terms of reproducibility, hoping for more vigorous support for our paper.
>
> ---
>
> **Response 1: Further implementation details.**
>
> First, the general input prompt of LLaMA-VQA is provided in Tab. 1 of the main paper. Specifically, the table represents the prompt for VQ → A, where the prompt from `[SOS]` to `Answer:` serves as a prefix and the model needs to generate the tokens after `Answer:`. Similarly, for VA → Q, `Video: ...`, `Choices: ...`, and `Answer: ...` tokens serve as prefix tokens and onwards is for a generation. The same manner is applied to QA → V.
>
> Also, LLaMA-VQA 7B is composed of 32 heads and 32 layers and the feature dimension $D$ is 4096. The total number of vocabularies used is 32,000 from SentencePiece Tokenizer. For visual encodings, each frame is resized by 224 $\times$ 224 and fed into CLIP VIT-L/14 to extract frame features.
>
> The hyperparameters of our main experiments (Tab.2 of the main paper) are shown in the table below.
> |  | Learning Rate | # adapter tokens ($N_p$) | Batch Size | Weight Decay | Epoch | Warmup Epoch | # frames ($N_v$) | Max Sequence Length ($N_v + N_q + N_a$) |
> | --- | --- | --- | --- | --- | --- | --- | --- | --- |
> | NExT-QA | 0.0225 | 10 | 64 | 0.16 | 5 | 2 | 10 | 128 |
> | STAR | 0.0175 | 10 | 64 | 0.16 | 5 | 2 | 10 | 128 |
> | DramaQA | 0.0225 | 10 | 64 | 0.16 | 5 | 1 | 10 | 384 |
> | VLEP | 0.02 | 10 | 64 | 0.25 | 5 | 2 | 10 | 256 |
> | TQVA | 0.00875 | 10 | 32 | 0.02 | 5 | 2 | 10 | 512 |
>
> We will include further implementation details in our final version and release our code if accepted.

---

### Meta-Review · Area_Chair_E6De · 2023-09-22

**Recommendation:** 4

**Metareview:**

The reviewers are unanimous that paper is sound and exciting. There is consensus on the quality of the evaluation and the proposed Flipped-VQA method as a sample effient means to train model and help understand their causal and temporal reasoning.

---

### Decision · Program_Chairs · 2023-10-07

**Decision:**

Accept-Main

**Comment:**

The reviewers are unanimous that paper is sound and exciting. There is consensus on the quality of the evaluation and the proposed Flipped-VQA method as a sample effient means to train model and help understand their causal and temporal reasoning.